# Experimental Predictions for Norm-Conserving Spontaneous Collapse

**DOI:** 10.3390/e25111489

**Published:** 2023-10-27

**Authors:** D. W. Snoke, D. N. Maienshein

**Affiliations:** 1Department of Physics and Astronomy, University of Pittsburgh, 100 Allen Hall, Pittsburgh, PA 15260, USA; 2Department of Mathematics, University of Pittsburgh, 301 Thackeray Hall, Pittsburgh, PA 15260, USA; dnm48@pitt.edu

**Keywords:** quantum field theory, quantum foundations, spontaneous collapse

## Abstract

Previous work has shown that nonlocal collapse in quantum mechanics can be described by a deterministic, non-unitary operator added to the standard Schrödinger equation. In terms of key aspects, this term differs from prior work on spontaneous collapse. In this paper, we discuss the possible predictions of this model that can be tested by experiments. This class of collapse model does not intrinsically imply unique experimental predictions, but it allows for the possibility.

## 1. Introduction

Spontaneous collapse theories of quantum mechanics have the appeal that they allow the experimental results of quantum mechanics to be accounted for within a mathematical formalism that needs no separate category of “measurement”, which is left off the books. It has also been argued that a consistent approach to gravitation requires some form of spontaneous collapse [1,2,3]. There have been various approaches to spontaneous collapse, including early work by Pearle [4] and Gisin [5,6], the Ghirardi-Rimini-Weber (GRW) model [7], a formalism that puts the GRW model into a relativistic framework [8], and models of continuous spontaneous localization (CSL) [9,10,11,12,13,14]. A common characteristic of these approaches is to argue that the equations of quantum mechanics must be altered to include new, non-unitary terms. This means that they are, at least in principle, experimentally falsifiable, and conversely, there can be predictions for experiments that give evidence in support of them.

The above models involve, in one way or another, a universal source of random fluctuations that interacts with all particles, even those in the vacuum of free space. This randomness is expected to lead to both the washing out of interference and other decoherence effects [15,16,17,18,19] and other ways of detecting noise [20,21,22,23]. So far, experiments have not detected this type of noise directly. In general, these models imply the nonconservation of energy that can be detected as heat because random spatial translations of the particles amount to a heat source [7]. This prediction has recently been tested with null results [24]. An alteration of the GRW model may allow for the dissipation of this extra heat [25].

Recent work [26,27] has presented a new approach inspired by these spontaneous collapse models but without an extrinsic noise term that would wash out interference in a vacuum. This model proposes a specific new term to be added to the Schrödinger equation, which gives the Born rule for measurements but also, as discussed below, conserves energy and the total particle number and conserves the norm of the full many-body wave function. The overall approach is entirely in the context of many-body quantum field theory without committing to a definite ontology of “particles”. All that is needed is to define the localized eigenstates of fermion fields.

Since total energy is conserved in this model, the question remains whether it gives any experimentally testable predictions. In the following, we show that this class of model *may* allow for novel experimental predictions but *need* not. In the absence of any unique experimental predictions, this approach would still have the appeal of internal self-consistency and agreement with known experiments.

## 2. Properties of the Non-Unitary Term

The non-unitary term added to the Hamitonian in the Schrödinger equation in the model derived in Ref. [27] is
(1)$$\begin{array}{ccc} \hat{V} & = & \sum_{n}i\hbar \omega_{R,n}\left(\langle {\hat{N}}_{n}\rangle \left(1-{\hat{N}}_{n}\right)-\left(1-\langle {\hat{N}}_{n}\rangle \right){\hat{N}}_{n}\right)\\ & = & \sum_{n}i\hbar \omega_{R,n}\left(\langle {\hat{N}}_{n}\rangle -{\hat{N}}_{n}\right), \end{array}$$
where ${\hat{N}}_{n}$ is the number operator acting on a state, *n*, which is a single-particle state within a general fermionic many-particle state. As shown in Ref. [27], this operator gives a Rabi rotation on the Bloch sphere between two states, namely a fermion that does not exist in state *n* (state “0”) and a fermion existing in state *n* (state “1”), and the scalar factor $\omega_{R,n}$ is the effective Rabi frequency. This factor is assumed to fluctuate randomly, dependent on local environmental fluctuations, as discussed further below. This model is taken to apply only to fermions, which naturally have two states corresponding to occupation 0 and 1, and does not apply to bosons (One might conjecture an analogous term for bosons but of the opposite nature, of the form $\langle {\hat{\alpha }}_{n}^{†}\rangle -{\hat{\alpha }}_{n}$, where ${\hat{\alpha }}_{n}=a_{n}^{†}+a_{n}$ is the boson wave amplitude, which would favor coherent states over Fock states and give the spontaneous appearance of classical waves. For another approach, see Ref. [28]).

Because this operator gives pure Rabi rotations, it conserves the norm of any many-body state. We can see this in the following calculation. Suppose that the total Hermitian Hamiltonian is given by $H=H_{0}+{\hat{V}}_{I}$, where $H_{0}$ defines the eigenstates of the system, and ${\hat{V}}_{I}$ is the full set of standard Hermitian interactions. We write the time evolution of the norm of the state using the *S*-matrix expansion for the Hermitian part of the Hamiltonian (see, e.g., Ref. [29], Section 8.1):(2)$$\begin{array}{ccc} d\langle \psi |\psi \rangle & = & \langle \psi_{t}|\psi_{t}\rangle -\langle \psi_{0}|\psi_{0}\rangle \\ & = & \langle \psi_{0}|e^{-iH_{0}t/\hbar }\left(1+\frac{i}{\hbar }\int_{0}^{t}\left({\hat{V}}_{I}\left(t^{\prime }\right)+{\hat{V}}^{†}\left(t^{\prime }\right)\right)dt^{\prime }\right.\\ & & \left.-\frac{1}{\hbar^{2}}\int_{0}^{t}\left({\hat{V}}_{I}\left(t^{\prime }\right)+{\hat{V}}^{†}\left(t^{\prime }\right)\right)dt^{\prime }\int_{0}^{t^{\prime }}\left({\hat{V}}_{I}\left(t^{\prime \prime }\right)+{\hat{V}}^{†}\left(t^{\prime \prime }\right)\right)dt^{\prime \prime }+\ldots \right)\\ & & \times \left(1-\frac{i}{\hbar }\int_{0}^{t}\left({\hat{V}}_{I}\left(t^{\prime }\right)+\hat{V}\left(t^{\prime }\right)\right)dt^{\prime }\right.\\ & & \left.-\frac{1}{\hbar^{2}}\int_{0}^{t^{\prime }}\left({\hat{V}}_{I}\left(t^{\prime }\right)+\hat{V}\left(t^{\prime }\right)\right)dt^{\prime }\int_{0}^{t^{\prime }}\left({\hat{V}}_{I}\left(t^{\prime \prime }\right)+\hat{V}\left(t^{\prime \prime }\right)\right)dt^{\prime \prime }+\ldots \;\right)e^{iH_{0}t/\hbar }|\psi_{0}\rangle \\ & & -\langle \psi_{0}|\psi_{0}\rangle , \end{array}$$
where for any operator, we write, $\hat{O}$, $\hat{O}\left(t\right)\equiv e^{iH_{0}t/\hbar }\hat{O}e^{-iH_{0}t/\hbar }$.

The set of all terms with just Hermitian ${\hat{V}}_{I}\left(t\right)$ operators must give unitary evolution if the entire wave function is taken into account. However, If the wave function is taken to refer only to a restricted subsystem, then the second-order expansion gives dissipative terms (as shown in Ref. [29], Section 8.1), which corresponds to the decay of the norm of the wave function; if the same second-order terms are used to compute the time-evolution of the density matrix, they give rise to the Lindbladian operator (see, e.g., Ref. [30], Section 19.3).

Moving on to those terms with the products of ${\hat{V}}_{I}\left(t\right)$ and $\hat{V}\left(t\right)$, we note that
(3)$$\begin{array}{ccc} \hat{V}\left(t\right) & = & \sum_{n}\left(\langle {\hat{N}}_{n}\rangle -e^{iH_{0}t/\hbar }{\hat{N}}_{n}e^{-iH_{0}t/\hbar }\right)=\hat{V}\left(0\right), \end{array}$$
since ${\hat{N}}_{n}$ commutes with $H_{0}$. Therefore, integrals over $\hat{V}\left(t\right)$ will be proportional to *t*, and terms with the products of integrals of ${\hat{V}}_{I}\left(t\right)$ and $\hat{V}\left(t\right)$ will be proportional to $t^{2}$, and therefore, will vanish for a small *t*, which is the differential limit taken here.

Finally, we turn to the terms with only a non-Hermitian $\hat{V}\left(t\right)$. The products of integrals of these will be proportional to $t^{2}$ or higher, which will vanish according to the same argument given above. We are left with just the linear terms:(4)$$\begin{array}{ccc} \langle \psi_{t}|\psi_{t}\rangle -\langle \psi_{0}|\psi_{0}\rangle & = & \frac{i}{\hbar }t\langle \psi_{0}|\left({\hat{V}}^{†}-\hat{V}\right)|\psi_{0}\rangle \\ & = & \sum_{n}2\omega_{R,n}t\langle \psi_{0}|\left(\langle {\hat{N}}_{n}\rangle -{\hat{N}}_{n}\right)|\psi_{0}\rangle \\ & = & \sum_{n}2\omega_{R,n}t\left(\langle {\hat{N}}_{n}\rangle -\langle {\hat{N}}_{n}\rangle \right)=0. \end{array}$$
We can, therefore, call the term (1) “quasi-unitary”—although it is not Hermitian, it is strictly norm-conserving, and if $\omega_{R,n}$ fluctuates randomly as positive and negative, then the time average of the adjoint of the operator is the same as the time average of the operator.

The form of (1) is similar to the terms in various other proposals in the literature, e.g., Refs. [5,11,12]. It is often assumed, based on the arguments discussed further in Appendix A, that a term like this must give a mechanism for superluminal communication. The general form of the argument is (1) to construct a density matrix for the state of interest and (2) then average this over many instances to get the density matrix for a mixed state that depends on a specific choice of measurement of one degree of freedom of a nonlocally entangled state, (3) act on this density matrix for with the non-Hermitian operator, and (4) from this, show that changes in the measurement can be detected nonlocally by changes in the density matrix.

This argument can only work, however, if the action of the non-Hermitian operator is the same for every instance in the average, which is carried out for the density matrix of a mixed state. If the non-Hermitian operator acts differently every time, namely with random fluctuations that lead to the Born rule, as discussed in Section 3, then its action is exactly the same as a standard quantum measurement, which cannot give superluminal communication. To put it another way, the calculation for the average evolution must take two limits in the proper order, and incorrect results are obtained if they are not taken in the proper order. One must *first* take the limit dt→0 to obtain the proper differential equation for any single time-evolution for one particular random walk, and then only *afterward* take the limit of an infinite number of random walks in an ensemble to get the average behavior.

Consider the entangled state of a system in which the fermion is in a superposition of both occupying and not occupying a state, *n*,
(5)$$\begin{array}{c} |\psi \rangle =\alpha_{n}|\psi_{0}\rangle |0\rangle +\beta_{n}|\psi_{1}\rangle |1\rangle , \end{array}$$
where |0〉 and |1〉 represent the unoccupied and occupied localized fermion state, *n*, of interest, and $|\psi_{0}\rangle$ and $|\psi_{1}\rangle$ represent the associated many-body states of the entire rest of the system (which may be large superpositions of many-body states). The states $|\psi_{0}\rangle$ and $|\psi_{1}\rangle$ are orthonormal because they have different total numbers of fermions, and the total fermion number is conserved. For norm-conserving evolution, the coefficients $\alpha_{n}$ and $\beta_{n}$ must always be normalized according to $\sqrt{|\alpha_{n}{|}^{2}+{\left|\beta_{n}\right|}^{2}}=1$. As discussed in Section 3, various physical processes can lead to time-dependent fluctuations of $\omega_{R,n}$, and these fluctuations of $\omega_{R,n}$ lead to a random walk of the coefficients $\alpha_{n}$ and $\beta_{n}$, which has the end result of either α or β becoming nearly exactly 1 while the other becomes nearly exactly 0, with the probability of each given by the Born rule. This has the same end result as a projection onto one state followed by the normalization of the wave function, as in standard measurement theory. Therefore, this type of random walk *cannot* produce the results that a standard Born-rule measurement does not produce.

The projection of a state onto one of two localized fermion states is sufficient to give nonlocal correlation. Consider the entangled state
(6)$$\begin{array}{c} |\Psi \rangle =\alpha_{1}\left|0,1\rangle \right|\psi_{1}\rangle +\alpha_{2}\left|1,0\rangle \right|\psi_{2}\rangle , \end{array}$$
where the 0s and 1s represent the fermion occupation numbers of two states, which can be widely spatially separated, and $|\psi_{1}\rangle$ and $|\psi_{2}\rangle$ are the associated environments (again, these may be highly complicated, massive superpositions). Then, according to the model proposed here, fluctuations in the local environment of each of these states leads to a random walk of $\alpha_{n}$ and $\beta_{n}$ for each electron state *n* via the term (1). Because the term acts on electrons in each of the atoms, when one electron state experiences an “upward” kick toward occupation $N_{n}=1$, the other experiences a downward kick toward zero occupation. Eventually, this correlated random walk will end with one of the states having an occupation of 1, and the other having an occupation of 0, effectively projecting the full state onto one of the two states in (Equation 6). This argument can be generalized to any number of spatially separated states in a superposition.

**Conservation rules.** While the term (1) is not Hermitian, we can derive several conservation rules for it. First, it conserves the total number of fermions in the system. This can be seen by computing the time evolution of the expectation value of the total number operator $\hat{N}=\sum {\hat{N}}_{n}$ predicted by the Schrödinger equation. By the same arguments as above, we keep only the terms that are linear in *t*, which gives, for an infinitesimal time lapse, *t*,
(7)$$\begin{array}{ccc} \langle \psi_{t}|\hat{N}|\psi_{t}\rangle -\langle \psi_{0}|\hat{N}|\psi_{0}\rangle & = & \langle \psi_{0}|\left(1+\frac{i}{\hbar }{\hat{V}}^{†}dt\right)\hat{N}\left(1-\frac{i}{\hbar }\hat{V}dt\right)|\psi_{0}\rangle -\langle \psi_{0}|\hat{N}|\psi_{0}\rangle \\ & ≃ & \frac{i}{\hbar }dt\langle \psi_{0}|\left({\hat{V}}^{†}\hat{N}-\hat{N}\hat{V}\right)|\psi_{0}\rangle \\ & = & \frac{i}{\hbar }Ndt\langle \psi_{0}|\left({\hat{V}}^{†}-\hat{V}\right)|\psi_{0}\rangle , \end{array}$$
where in the last line, we have used the property of $|\psi_{0}\rangle$, meaning that it is a superposition of states all with the same total number of fermions, *N*, since any standard Hermitian interaction that changes the states of the individual fermions conserves the total fermion number. The change in $\langle \hat{N}\rangle$ is, therefore, 0 when using the results of (Equation 4).

This term also conserves the total energy of the system under the same conditions evoked for energy conservation when deriving irreversible behavior under Fermi’s golden rule and the quantum Boltzmann equation (see, e.g., Sections 4.7 and 4.8 of Ref. [29]). In this case, we can calculate the time dependence of the expectation value of $H_{0}=\sum \hbar \omega_{n}{\hat{N}}_{n}$. Analogous to (Equation 7) above, we have
(8)$$\begin{array}{ccc} & & \langle \psi_{t}|H_{0}|\psi_{t}\rangle -\langle \psi_{0}|H_{0}|\psi_{0}\rangle =\frac{i}{\hbar }dt\langle \psi_{0}|\left({\hat{V}}^{†}H_{0}-H_{0}\hat{V}\right)|\psi_{0}\rangle \\ & & \;=\hbar dt\sum_{n,n^{\prime }}\omega_{R,n}\omega_{n^{\prime }}\langle \psi_{0}|\left(\left(\langle {\hat{N}}_{n}\rangle -{\hat{N}}_{n}\right){\hat{N}}_{n^{\prime }}+{\hat{N}}_{n^{\prime }}\left(\langle {\hat{N}}_{n}\rangle -{\hat{N}}_{n}\right)\right)|\psi_{0}\rangle \\ & & \;=\hbar dt\sum_{n,n^{\prime }}\omega_{R,n}\omega_{n^{\prime }}\left(2\langle {\hat{N}}_{n}\rangle \langle {\hat{N}}_{n^{\prime }}\rangle -2\langle {\hat{N}}_{n}{\hat{N}}_{n^{\prime }}\rangle \right). \end{array}$$
This is equal to zero if $\langle {\hat{N}}_{n}{\hat{N}}_{n^{\prime }}\rangle$ factors to $\langle {\hat{N}}_{n}\rangle \langle {\hat{N}}_{n^{\prime }}\rangle$, which, as shown in Ref. [31], is equivalent to the strong decoherence limit, which is the same limit used to justify the quantum Boltzmann equation. The same approach can be taken if we add a number-conserving interaction to the Hamiltonian with terms that take the form $Aa_{i}^{†}a_{j}^{}$. In this case, we also find energy conservation as long as $\langle {\hat{N}}_{n}a_{i}^{†}a_{j}^{}\rangle =\langle {\hat{N}}_{n}\rangle \langle a_{i}^{†}a_{j}^{}\rangle$, which is also the case when there is strong decoherence.

## 3. Numerical and Mathematical Results Showing the Born Rule

Ref. [27] showed that for any single fermion state, *n*, adding the operator (1) to the Schrödinger equation maps to the motion of a vector on a Bloch sphere, given by $\vec{U}=\left(\sin\theta \cos\phi ,\sin\theta \sin\phi ,\cos\theta \right)$, representing the superposition (Equation 5) with $\alpha =e^{i\phi }\sin\theta /2$ and β=cosθ/2. The motion of this Bloch vector is then governed by the dynamical equations
(9)$$\begin{array}{ccc} \frac{\partial U_{2}}{\partial t} & = & -\omega_{R,n}U_{3}{\left(1-U_{3}^{2}\right)}^{1/2}\\ \frac{\partial U_{3}}{\partial t} & = & \omega_{R,n}U_{2}{\left(1-U_{3}^{2}\right)}^{1/2}. \end{array}$$
Here, the factor ${\left(1-U_{3}^{2}\right)}^{1/2}$ gives two attractors corresponding to $\langle {\hat{N}}_{n}\rangle =0$ and 1; the “collapse” of the wave function occurs when the Bloch vector hits either of these attractors. The dynamics of this Bloch vector model were analyzed numerically in Ref. [26]. When the Rabi factor $\omega_{R,n}$ fluctuates randomly in time, the Bloch vector undergoes a random walk between the two attractors.

The t→∞ results of many random walks of this type have been shown numerically to give the Born probability rule with high accuracy. Figure 1 shows a comparison of the linear prediction of the Born rule for the probability of collapse to the t→∞ results of a numerical calculation using this model, using a value of $\omega_{R,n}$ picked randomly for a sequence of time intervals dt, with the probability of the value within each time interval given by the Lorentzian (Cauchy) distribution
(10)$$\begin{array}{ccc} P(\omega_{R,n}) & = & \frac{1}{\pi }\frac{\gamma }{\omega_{R,n}^{2}+\gamma^{2}}, \end{array}$$
where γ gives the characteristic range of the fluctuations, with γdt≪1 and a cutoff of γdt≤0.5. This is the expected distribution for a wide range of typical processes, in which the fluctuations have an exponentially decaying correlation function in time, i.e., no long-time correlation (see, e.g., Refs. [32,33]). The Wiener-Khinchtine theorem says that the frequency spectrum of a classical noise source is given by the Fourier transform of the temporal correlation function (see, e.g., Ref. [29], Section 9.5). For an exponentially decaying temporal correlation function $C\left(t\right)\propto e^{-\gamma |t|}$, this implies
(11)$$\begin{array}{ccc} F\left(\omega \right)=\frac{1}{2\pi }\int_{-\infty }^{\infty }e^{-\gamma |t|}e^{i\omega t}dt & = & \frac{1}{2\pi }\int_{-\infty }^{0}e^{\gamma /\tau }e^{i\omega t}dt+\frac{1}{2\pi }\int_{0}^{\infty }e^{-t\gamma }e^{i\omega t}dt\\ & = & \frac{1}{\pi }\frac{\gamma }{\gamma^{2}+\omega^{2}}. \end{array}$$
Note that the function $e^{-\gamma |t|}$ has a discontinuity in its slope at t=0. The Fourier transform of this gives infinitely high-frequency components in the spectral function, which are nonphysical. Realistically, this sharp discontinuity should be replaced by a smoothly curved peak at t=0 over some small time range. This was accounted for in the numerical model by a high-frequency cutoff in the Lorentzian distribution.

The same linear result was found in numerical simulations using a Gaussian distribution of fluctuations of $\omega_{R,n}$. The Born rule holds for all of the random walks considered numerically on a finite interval in the t→∞ limit, which, mathematically, is because these walks are “martingale” walks. Martingales are defined in Appendix B, where it is shown that the numerical simulations will converge to +1 or −1 with probabilities linearly proportional to their starting position.

## 4. Quasi-Unitary Evolution Is the Same As Weak Measurement

As discussed in Section 2, the nonlinear operator (Equation 1) is similar in form to that suggested by Gisin [5] and others, and it has been argued that the nonlinear terms of this type intrinsically imply the possibility of superluminal communication. As shown in Section 2, however, the operator (Equation 1) is strictly norm-conserving, so that the Bloch vector for the full wave function (Equation 5) always has an exact unit length. This Bloch vector represents the full density matrix for the system as well as the environment that is entangled with it. When this entanglement with the environment is taken into account, the density matrix for any single random walk is always a pure state of the full many-body wave function, with a Bloch vector of unit length. As discussed in Section 1, in the t→∞ limit, a random walk of the Bloch vector implies that both the system and the environment collapse into one of two possibilities in accordance with the Born rule.

In Section 2, we asserted that the average of many random walks using the operator (Equation 1) will always give an evolution of the density matrix that is consistent with the standard Born rule of measurements in quantum mechanics. This can be seen explicitly by calculating the prediction of a random walk (“quantum trajectory”) using standard quantum mechanics in the “weak measurement” limit. Weak measurement theory [34,35,36,37] (for reviews, see, e.g., Refs. [38,39,40]) can be viewed as a type of spontaneous collapse, but it does not invoke any explicit non-unitary terms; instead, many weak Copenhagen-type measurements are assumed to occur, without specifying the observer. Recent work [41] has shown that weak measurement theory can be integrated into the decoherence approach of Zurek. Zurek’s approach (e.g., Ref. [42]) shows that the collapse of a density matrix to the diagonal can occur via unitary dynamics (sometimes called “first collapse,”) but cannot reproduce the statistics of individual quantum trajectories, i.e., single measurements (“second collapse”). Therefore, weak measurement theory is completely compatible with standard quantum mechanics but gives no insight into the cause of collapses. In this section, we show that weak measurement gives exactly the same operator as posited in (Equation 1).

Consider the following scenario of weak measurement, following the model from Ref. [43]. At time t=0, the initial state is
(12)$$\begin{array}{c} |\psi \rangle =|\phi \rangle |X=0\rangle , \end{array}$$
where |X〉 is the state of an external detector with a center-of-mass value at *X* (e.g., the position of a needle in a meter), and |ϕ〉=α|−1〉+β|1〉 is the internal state, which is in a superposition. The external state, in general, has position uncertainty in a Gaussian
(13)$$\begin{array}{c} |X\rangle =\int dx\;e^{-{\left(x-X\right)}^{2}/2\sigma^{2}}|x\rangle , \end{array}$$
where $|x_{0}\rangle$ is the state of the external detector at exactly $x_{0}$.

At t=0, a weak interaction of the system with the detector is turned on briefly, taking the form of
(14)$$\begin{array}{c} \hat{V}=-gS_{z}P, \end{array}$$
where $S_{z}$ is the standard spin operator, P=−iℏ∂/∂x is the momentum operator of the external center of mass *X*, and *g* is some small number. After a short time dt, the state of the system is
(15)$$\begin{array}{ccc} |\psi^{\prime }\rangle & = & |\psi \rangle +gdt\int dx|x\rangle \left(-\alpha \frac{\partial }{\partial x}e^{-x^{2}/2\sigma^{2}}|-1\rangle +\beta \frac{\partial }{\partial x}e^{-x^{2}/2\sigma^{2}}|1\rangle \right)\\ & = & |\psi \rangle +gdt\int dx|x\rangle \left(\alpha \frac{x}{\sigma^{2}}e^{-x^{2}/2\sigma^{2}}|-1\rangle -\beta \frac{x}{\sigma^{2}}e^{-x^{2}/2\sigma^{2}}|1\rangle \right)\\ & = & \int dx|x\rangle e^{-x^{2}/2\sigma^{2}}\left(\alpha \left(1+\frac{xgdt}{\sigma^{2}}\right)|-1\rangle +\beta \left(1-\frac{xgdt}{\sigma^{2}}\right)|1\rangle \right)\\ & ≃ & \int dx|x\rangle \left(\alpha e^{-{\left(x-gdt\right)}^{2}/2\sigma^{2}}|-1\rangle +\beta \;e^{-{\left(x+gdt\right)}^{2}/2\sigma^{2}}|1\rangle \right). \end{array}$$
The last line is commonly used in discussing this scenario, but we will stick with the third line.

We now do a strong measurement of the external detector to collapse it to a definite value $x=x_{0}$. By defining $df=xgdt/\sigma^{2}\ll 1$, the state of the whole system is then
(16)$$\begin{array}{ccc} |\psi^{\prime \prime }\rangle & = & \frac{1}{\sqrt{{\left|\alpha \right|}^{2}{\left(1+df\right)}^{2}+{\left|\beta \right|}^{2}{\left(1-df\right)}^{2}}}\left(\alpha (1+df)|-1\rangle +\beta (1-df)|1\rangle \right)|x_{0}\rangle \\ & ≃ & \frac{1}{\sqrt{1+{2|\alpha |}^{2}df-2{\left|\beta \right|}^{2}df}}\left(\alpha (1+df)|-1\rangle +\beta (1-df)|1\rangle \right)|x_{0}\rangle \\ & ≃ & \left(1-{\left|\alpha \right|}^{2}df+{\left|\beta \right|}^{2}df\right)\left(\alpha (1+df)|-1\rangle +\beta (1-df)|1\rangle \right)|x_{0}\rangle . \end{array}$$
The change in time is then
(17)$$\begin{array}{ccc} d|\psi \rangle =|\psi^{\prime \prime }\rangle -|\psi \rangle & = & gdtx_{0}\left({\alpha (1-|\alpha |}^{2}+{\left|\beta \right|}^{2}{)|-1\rangle -\beta (1+|\alpha |}^{2}-{\left|\beta \right|}^{2})|1\rangle \right)|x_{0}\rangle \\ & = & gdtx_{0}\left((|\beta {|}^{2}-|\alpha {|}^{2})(\alpha |-1\rangle +\beta |1\rangle )-(-\alpha |-1\rangle +\beta |1\rangle \right)|x_{0}\rangle \\ & = & gdtx_{0}\left(\langle S_{z}\rangle -S_{z}\right)\left|\phi \rangle \right|x_{0}\rangle . \end{array}$$
We have, thus, obtained a nonlinear operator of exactly the same form as (Equation 1) because the $S_{z}$ operator has exactly the same operation as ${\hat{N}}_{n}$ acting on two states; the nonlinearity arises from the fact that a standard quantum measurement is an intrinsically non-unitary process.

If σ is large, then $x_{0}$ is equally probable to be positive or negative. Therefore, if we apply the above process of weak measurement multiple times, with each time allowing the position of the needle in the detector to gain uncertainty due to normal wavepacket spreading, and then carrying out a measurement of the classical center of mass of the needle each time, we will get a random walk that is exactly equivalent to the results discussed in Section 2.

The significance of this mapping of weak measurement theory to the proposed non-Hermitian operator (Equation 1) for spontaneous collapse is that because weak measurement theory is based on standard measurement theory, the random walk with the non-Hermitian operator (Equation 1) *cannot* give any effects that violate known physics; in particular, the term (Equation 1) does not intrinsically allow superluminal communication or lead to a violation of energy conservation because weak measurement does not, and the two are mathematically identical. Due to environmental fluctuations, the random walk will give a density matrix corresponding to a non-unit-length Bloch vector only when the results of many trials in an ensemble are averaged; for any single random walk, it will always correspond to a Bloch vector of unit length.

Therefore, we can switch our viewpoint and adopt the operator (Equation 1) as a fundamental postulate. By the martingale rule discussed in Appendix B, the result of many sequential weak measurements will obey the Born rule, which is to say, many weak measurements will give a strong measurement, which makes sense since the same information is extracted, whether quickly or slowly. Thus, we can postulate that every strong measurement is actually the end result of many weak measurements of the form (Equation 1), which are intrinsic, rather than needing to rig the particular scenario discussed above.

This result depends crucially, however, on the assumption that the steps are random. In the case of weak measurement theory, this was true because of the axiomatic assumption of the Born rule for measurements. In the case of the operator (Equation 1), as a new term in the Schrödinger equation, the randomness comes from the randomness of $\omega_{R,n}\left(t\right)$. This, in turn, depends on what physical processes we believe affect $\omega_{R,n}\left(t\right)$. If $\omega_{R,n}\left(t\right)$ is nonrandom and deterministically predictable, then the model presented in Section 2 *does* imply the possibility, at least in principle, of superluminal communication, as we will see below.

## 5. Possible Sources of Fluctuations and Experimental Implications

Let us now consider three different approaches to the physical source of the fluctuations of $\omega_{R,n}$ in this model.

**Universal fluctuations from a novel source.** The first possibility we consider is that the fluctuations in $\omega_{R,n}$ are unrelated to the local environment of the state *n*, and come from some universal source, such as gravity noise, dark matter, or some other fundamental field. This noise source would presumably still act locally in spacetime but would not be directly related to the presence of regular matter or energy. In this case, $\omega_{R,n}\left(t\right)$ would be fundamentally unpredictable, and no superluminal communication would be possible. This approach has much in common with the GRW and related Penrose-Diósi models—it predicts the decoherence and collapse that are unrelated to the presence of any detector; particle states decohere in a pure vaccum on some length and time scale. However, it does not imply the nonconservation of energy; as is shown in Section 2, energy is conserved under the same conditions of decoherence as Fermi’s golden rule.

**Local fluctuations of the environment.** A second possibility, proposed in Refs. [26,27], is that $\omega_{R,n}$ is directly the result of fluctuations of the local environment. For example, the fluctuation of the local energy density, which gives the fluctuation of the phase precession of state *n*, could give this. This has the appeal that measurement and collapse are directly related to the decoherence found in detectors and in any macroscopic system with strong decoherence.

In this case, we can posit that the full operator, which gives the nonlinear correction to the Schrödinger equation, is
(18)$$\begin{array}{c} \hat{V}\left(t\right)=i\xi \int d^{3}r\frac{\partial }{\partial t}\langle H\left(\vec{r},t\right)\rangle \left(\langle \Psi^{†}\left(\vec{r}\right)\Psi \left(\vec{r}\right)\rangle -\Psi^{†}\left(\vec{r}\right)\Psi \left(\vec{r}\right)\right), \end{array}$$
where $\Psi^{†}\left(\vec{r}\right)$ and $\Psi (\vec{r})$ are spatial field operators (cf. Ref. [29], Section 4.6), $H(\vec{r},t)$ is the standard energy operator from unitary physics, and ξ is a small parameter with units of time, which is a new physical constant. A fully relativistic version of this term is discussed in Appendix C.

The term (Equation 18) is equivalent to the form (1) under the assumption of coarse graining; that is, if we assume that $\partial /\partial t\langle H\left(\vec{r},t\right)\rangle$ is slowly varying in space for scales of length that are large compared to the localized states, *n*. In that case, we can write
(19)$$\begin{array}{c} \hat{V}=i\xi \sum_{\vec{R}}\frac{\partial }{\partial t}\langle H\left(\vec{R},t\right)\rangle \int_{V(\vec{R})}d^{3}r\left(\langle \Psi^{†}\left(\vec{r}\right)\Psi \left(\vec{r}\right)\rangle -\Psi^{†}\left(\vec{r}\right)\Psi \left(\vec{r}\right)\right), \end{array}$$
where $\vec{R}$ gives the positions of the coarse grains, and the integral is over the volume inside a coarse grain. We can then write the Fourier series
(20)$$\Psi^{†}\left(\vec{r}\right)=\sum_{n}\phi_{n}^{*}\left(\vec{r}\right)a_{n}^{†},$$
where $\phi_{n}\left(\vec{r}\right)$ is the wave function, and $a_{n}^{†}$ is the creation operator for an eigenstate *n*, assuming that these eigenstates are localized within a coarse grain. Then, (Equation 19) becomes
(21)$$\begin{array}{c} \hat{V}=i\xi \sum_{\vec{R}}\frac{\partial }{\partial t}\langle H\left(\vec{R},t\right)\rangle \sum_{n,n^{\prime }}\int_{V(\vec{R})}d^{3}r\;\phi_{n}^{*}\left(\vec{r}\right)\phi_{n^{\prime }}\left(\vec{r}\right)\left(\langle a_{n}^{†}a_{n^{\prime }}^{}\rangle -a_{n}^{†}a_{n^{\prime }}^{}\right). \end{array}$$
When using the orthogonality relation $\int d^{3}r\phi_{n}^{*}\left(\vec{r}\right)\phi_{n^{\prime }}\left(\vec{r}\right)=\delta_{n,n^{\prime }}$ then this gives us
(22)$$\begin{array}{c} \hat{V}=i\xi \sum_{\vec{R},n}\frac{\partial }{\partial t}\langle H\left(\vec{R},t\right)\rangle \left(\langle a_{n}^{†}a_{n^{\prime }}^{}\rangle -a_{n}^{†}a_{n^{\prime }}^{}\right), \end{array}$$
which is equivalent to (1) under the assumption that the states, *n*, of interest are localized.

If the local environment has fluctuating energy density, then this will give the type of random walk discussed in Section 2, which reproduces the Born rule. Typical values of fluctuations in atom-based detectors are meV per picosecond, and the numerical results discussed above indicate that the order of 100 steps in a random walk is required to have a “collapse”; therefore, we can estimate ξ is of the order of $10^{-11}$ s/eV. This form does not violate energy conservation or produce heat because the fluctuations are driven entirely by real energy fluctuations that exist in the local environment.

With the form (Equation 19), however, we have the possibility of the external control of the detection statistics. Suppose that instead of waiting for local environmental fluctuations, we control ∂〈H〉/∂t directly, for example, when using an intense laser pulse to give it a constant value that overcomes any local fluctuations. In that case, at least in principle, we can “jam” the detector locally to always detect a particle that is in violation of the Born rule by producing a rapid increase in the local energy density. Then, any remote particles entangled with the one we have jammed will also experience a violation of the Born rule, which would allow superluminal communication. A person could monitor a local detector for deviations from the Born rule with an agreed-upon flux of entangled particles sent between the two detectors, and a deviation from the Born rule could be registered as one bit of information, e.g., a bit value of 1= for the Born rule, and a bit value of 0= for the non-Born rule.

For the estimated value of ξ above and the atomic states of the order of the Bohr radius, jamming the detection by using this method would require a rate of energy density change in the order of $10^{16}$ W/cm${}^{3}$. This would correspond to, for example, 10 kW of laser power absorbed in a volume with dimension of 1 micron. This is high but is within the range of modern technology.

**Nonjammable environmental fluctuations.** Suppose that the above experimental test is carried out and that no change from the Born-rule statistics is found in the entangled pairs. Does this falsify the proposed model? No, because it could be the case that form (Equation 1) is still correct but that the connection of $\omega_{R,n}$ to the rate of change in the local energy density is incorrect. The form (Equation 1) allows for a wide variety of theories for what physical processes lead to the Rabi rotations. For example, the Rabi frequency could be proportional to the second derivative in time, rather than the first derivative, which would still be sensitive to local fluctuations but would require an acceleration of the local energy density increase to give detectable “jamming.” It could also be the case that the Rabi rotations are induced by a term proportional to the product of local fluctuations and a global, universal noise source, as discussed above. In that case, there would still be no decoherence in a vacuum, but the background fluctuations would prevent any jamming of the detection. It is also simply possible that the effect of the local fluctuations is not linear but saturates, e.g., instead of ∂〈H〉/∂t, it is proportional to tan[∂〈H〉/∂t].

If no jamming is possible and no unique experimental evidence is found, this model is, therefore, not falsified, but it loses the appeal of unique predictions. In that case, the appeal of this spontaneous collapse model would primarily be that it carries less philosophical baggage, with a fairly plebian description of reality, entailing just the evolution of waves with resonances and nonlinearities without such things as parallel universes and the injection of human consciousness as a metaphysical entity while not violating any of the experimental results of standard quantum mechanics.

## 6. Conclusions

The above considerations show that experimental tests of spontaneous collapse, such as superluminal communication, are not automatically implied by the existence of a nonlinear term in the Schrödinger equation in the form (Equation 1), but are, in principle, allowed for some versions of the physical mechanism that gives the fluctuations.

As discussed in Ref. [27], the most natural way to preserve a “narrativity” in spontaneous collapse theories of this type is to posit a preferred reference frame, such as the rest frame of the cosmic microwave background (i.e., the rest frame of the center of mass of the universe). In this case, even in the case of superluminal communication, there will be no grandfather paradoxes because any return communications must occur in the same preferred reference frame. Appendix C gives further discussion on how to implement this type of collapse in a relativistic framework.

As discussed above, if no physical effect such as jamming is seen, it does not automatically falsify the proposed mechanism because there may be intrinsic physical properties that always give a noisy signal. In that case, the appeal of this mechanism is that it puts measurement into the realm of the standard kinematic descriptions of waves. What the present model shows is that a spontaneous collapse model that agrees with all the known experimental results and is as logically coherent as possible.

## Figures and Tables

**Figure 1 entropy-25-01489-f001:**
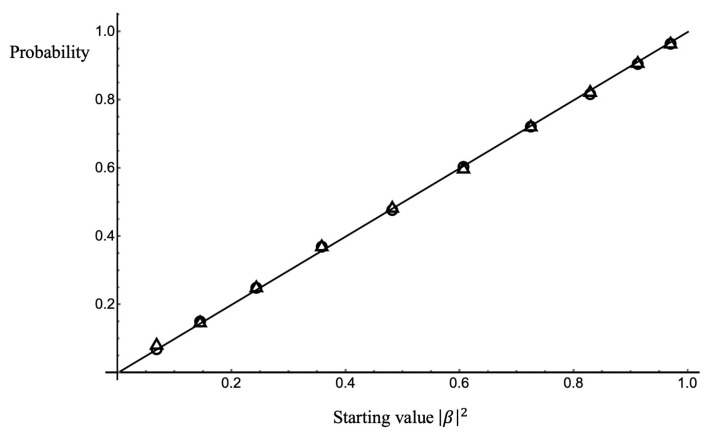
Comparison of the numerical results for the model with the Born rule (solid line). Circles: probability of end state $\langle {\hat{N}}_{n}\rangle =1$ for γdt=0.002. Squares: γdt=0.02. For each value of γdt, 8000 trials were run for each data point, and the number of steps in each trial was limited to 4000.

## Data Availability

The numerical model used for this study as well as Ref. [26] can be found at https://notebookarchive.org/2022-07-5la04jb.

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
