# Peer review of "Experimental Predictions for Norm-Conserving Spontaneous Collapse"

_entropy, 2023, doi:10.3390/e25111489_

Round 1
Reviewer 1 Report
The collapse of the wave function has been a controversial element of
quantum mechanics ever since the Copenhagen inyerpretation has been
postulated. On the one hand, it appears to be indispensable in order
to incorporate measurements as the prototypical interface between
quantum processes proper and the observable world following classical
rules. On the other hand, it is incompatible with the unitary time
evolution generated by the Schrödinger equation, the theoretical
framework of quantum theory that is generally approved and amply
corroborated by experimental evidence. Several attempts have been made
in the course of the debate to reconcile the collapse with unitary
quantum theory, each of which had to sacrifice the one or other
essential ingredient of the traditional Copenhagen view. An important
and remarkably successful approach interprets the collapse as a
manifestation of decoherence, induced by the coupling to an
environment modelled as a macroscopic many-body system such as the
electromagnetic field. In this case, however, the spontaneous collapse
is replaced by a cntinuous reduction taking place on a finite, if
short, timescale. (This account of the collapse could well be addressed as well in the introduction, for comparison and for the
completeness of the panorama.)
The present paper is dedicated to an alternative approach that seeks
preserving the spontaneous character of the collapse, but now on the
expense of the unitarity of the underlying time evolution.
Specifically, it adds a term to the Hamiltonian in the Schrödinger
equation, representing an additional component of the potential, that
lets the the total system tend towards an asymptotic state consistent
with the Born rule for the outcome of quantum measurements. It lacks
the Hermiticity otherwise required for quantum operators representing
observables and violates the nonlinearity of the Schrödinger equation,
but can nevertheless be shown to preserve the norm of the total wave
function.
While this model has already been introduced in previous publications,
the authors here consider its observability in experiments and in
particular the crucial question whether it would imply the possibility
of superluminal communication (which is excluded by Special Relativity
Theory and has been ruled out in experiments around the Einstein-
Podolsky-Rosen paradox and Bell's Theorem). Their central result is
that that such experimental consequences are not necessarily implied
by the model but can arise under plausible additional assumptions.
This is technically very sound work that provides a strong point in
favour of the model for spontaneous collapse the authors put forward.
A remarkable virtue which the authors do not even mention is that it
not only reproduces the loss of coherence of the measured system but
also its projection onto an eigenstate of the measured observable.
This return to a pure state again violating unitarity and is sometimes called the "second collapse" in the literature on quantum measurement.
Possible criticism therefore can only contrast spontaneous collapse
with competing approaches, in particular those based on decoherence
induced by an environment. Some principal arguments come to mind:
– The extra term, the authors' Eq. (1), in its relatively involved
form, appears to arise ad hoc, lacking a plausible foundation in the
physics of quantum measurement. To be sure, the form derived in Eq.
(17) for a specific scenario is quite convincing and supports the
general model (1). However, a similarly direct justification should
also be available for this general form.
– A striking feature of the model is that it implies the state vector
to approach one of two points in projective Hilbert space (in the case
of measurements of observables with more than two discrete eigenvalues, even up a numerably infinite number). More than one such
"attractor", as the authors call it, is not allowed by standard
quantum mechanics, but seems to be unavoidable in view of
measurement theory. It would therefore be desirable that the authors
dedicate more space to explaining how this effect comes about, more
than just stating the asymptotics of the random walk, as in the
paragraph following Eq. (3). The dynamics given in Eq. (9) is a
valuable hint, but an interpretation of the process in qualitative
terms would be useful.
– In fact, also the model discussed in this paper does not get along
without a random fluctuations and assumptions concerning its features
and its origin in quantum noise, gravity noise or other fancy sources.
In this respect, it is much closer to decoherence models of the
collapse than one might expect, which rises the question whether a
synthesis could be within reach.
In this context, I would like to mention that a unitary approach to
quantum measurement that reproduces the "first" and the "second"
collape, based on a discrete model of the environment, has been
published recently (my Ref. [1] below). Applying it to spin
measurement, the authors obtain similar results as reported in the
present manuscript, concerning the asymptotic approach to two
attractors (spin up vs. spin down) in the projective Hilbert space.
All in all, this paper can stimulate the discussion around the nature
of the collapse and certainly deserves being pulished in Entropy. To increase the impact of their work, the authors may find it helpful to
take the comments and suggestions, pointed out above, into account in
a final version.
[1] T. Dittrich, O. Rodríguez, C. Viviescas, "Simulating spin
measurement with a finite heat bath model for the environment",
Phys. Rev. A 106, 042203 (2022).
Reviewer 2 Report
The present work explores the consequences of a spontaneous collapse (SC) model for fermions formerly considered by the authors in Refs. 27 and 28 be one of the authors. Unlike other SC models formerly proposed in the literature, the main feature of the current one is that it conserves the total energy.
Although I am not in favor of SC models, mainly because there has no been so far any empirical evidence for them, neither there is a consistent theory behind (in general, all of them arise from ad hoc hypotheses rather than from a first-principle theory), I consider that they certainly contribute to keep alive and stir the debate on the quantum foundations, providing new conceptual frameworks to reconsider the problem of the reduction of the wavefunction. This is the case of the present work, grounded on an energy-preserving non-unitary term. There is a discussion on possible sources for the fluctuations described by the non-unitary operator, but, of course, they are just suitability conjectures with no empirical basis. This part, in my opinion, might be a bit misleading, but I understand that arguments go as far as a SC model can go.
Summing up, I consider that the present work is publishable in the current form, although it should be clearly stated that, so far, there is neither empirical evidence for the proposed model, nor there is a firm proposal for an experiment that can confirm (or refute) it. The basis for this statement is that, in several decades following the issue, I have not seen any significant advance.
As far as I have seen, there a few minor issues that should also be checked:
Lines 87-88: it should be “… its action is exactly the same as …” Please, remove “the” at the end of line 87.
Page 6, second paragraph of Sec. 4: in my opinion, it would be appropriate adding the original reference for the weak measurement as well as for the related concept of weak value.
Equation 7: in the second line, shouldn’t it be “\approx” instead of “=”?
Line 337: it should be “… the appeal of this mechanism …” Please, remove current “for” between “of” and “this”.
Line 341: it should be “ÄŒaslav” instead of “Caslov”
Overall, the quality of the English is fine. Nonetheless, I have detected a few issues that should be checked, as it is mentioned in the main report.
Reviewer 3 Report
See attached file.

The English is fine, aside from minor typos.
Round 2
Reviewer 3 Report
The questions and minor issues raised in the first referee report have been addressed. I believe the paper is suitable for publication in its present form.
Author Response
We thank the referee for this positive assessment.